# PROPOSAL-CONTRASTIVE PRETRAINING FOR OBJECT DETECTION FROM FEWER DATA

**Quentin Bouniot[1,2], Romaric Audigier[1], Angélique Loesch[1], Amaury Habrard[2,3]**
[1] Université Paris-Saclay, CEA, LIST, F-91120, Palaiseau, France
[2] Université Jean Monnet Saint-Etienne, CNRS, Institut d Optique Graduate School,
  Laboratoire Hubert Curien UMR 5516, F-42023, Saint-Etienne, France
[3] Institut Universitaire de France (IUF)
`firstname.lastname@{cea.fr, univ-st-etienne.fr}`

## ABSTRACT

The use of pretrained deep neural networks represents an attractive way to achieve strong results with few data available. When specialized in dense problems such as object detection, learning local rather than global information in images has proven to be more efficient. However, for unsupervised pretraining, the popular contrastive learning requires a large batch size and, therefore, a lot of resources. To address this problem, we are interested in transformer-based object detectors that have recently gained traction in the community with good performance and with the particularity of generating many diverse object proposals. In this work, we present Proposal Selection Contrast (ProSeCo), a novel unsupervised overall pretraining approach that leverages this property. ProSeCo uses the large number of object proposals generated by the detector for contrastive learning, which allows the use of a smaller batch size, combined with object-level features to learn local information in the images. To improve the effectiveness of the contrastive loss, we introduce the object location information in the selection of positive examples to take into account multiple overlapping object proposals. When reusing pretrained backbone, we advocate for consistency in learning local information between the backbone and the detection head. We show that our method outperforms state of the art in unsupervised pretraining for object detection on standard and novel benchmarks in learning with fewer data.

## 1 INTRODUCTION

In recent years, we have seen a surge in research on unsupervised pretraining. For some popular tasks such as Image Classification or Object detection, initializing with a pretrained backbone helps train big architectures more efficiently (Chen et al., 2020b; Caron et al., 2020; He et al., 2020). While gathering data is not difficult in most cases, its labeling is always time-consuming and costly. Pretraining leverages huge amounts of unlabeled data and helps achieve better performance with fewer data and fewer iterations, when finetuning the pretrained models afterwards.

The design of pretraining tasks for dense problems such as Object Detection has to take into account the fine-grained information in the image. Furthermore, complex object detectors contain different specific parts that can be either pretrained *independently* (Xiao et al., 2021; Xie et al., 2021; Wang et al., 2021a; Hénaff et al., 2021; Dai et al., 2021b; Bar et al., 2022) or *jointly* (Wei et al., 2021). The different pretraining possibilities for Object Detection in the literature are illustrated in Figure 1. A pretraining of the backbone tailored to dense tasks has been the subject of many recent efforts (Xiao et al., 2021; Xie et al., 2021; Wang et al., 2021a; Hénaff et al., 2021) (*Backbone Pretraining*), but few have been interested in incorporating the detection-specific components of the architectures during pretraining (Dai et al., 2021b; Bar et al., 2022; Wei et al., 2021) (*Overall Pretraining*). Among them, SoCo (Wei et al., 2021) focuses on convolutional architectures and pretrains the whole detector, *i.e.* the backbone along with the detection heads (approach *e.* in Figure 1), whereas UP-DETR (Dai et al., 2021b) and DETReg (Bar et al., 2022) pretrain only the transformers (Vaswani et al., 2017) in transformer-based object detectors (Carion et al., 2020; Zhu et al., 2021) and keep the backbone

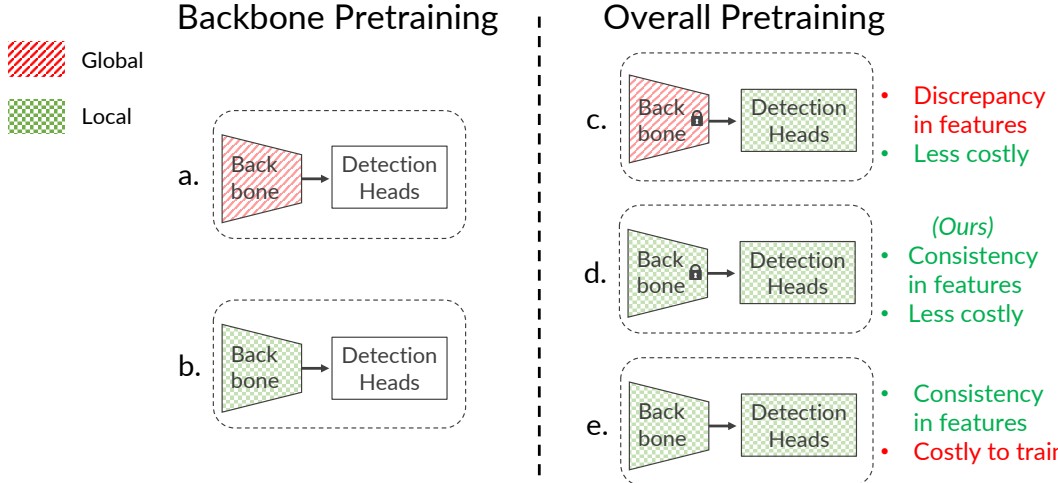

Figure 1: Illustration of the different pretraining possibilities for Object Detection. The pretraining can be either limited to the backbone (*left*), or overall including the detection heads (*right*). The few previous overall approaches either suffer from a discrepancy in the features between the backbone, that is trained at the image-level (*global*), and the detection heads, trained to encode object-level (*local*) information (*c.*), or from the cost of training lots of parameters with a large batch size (*e.*).

fixed (approach *c.* in Figure 1). Due to the numerous parameters that must be learned and the huge number of iterations needed because of random initialization, pretraining the entire detection model is expensive (Figure 1, *e.*). On the other hand, pretraining only the detection-specific parts with a fixed backbone leads to fewer parameters and allows leveraging strong pretrained backbones already available. However, fully relying on aligning embeddings given by the fixed backbone during pretraining and those given by the detection head, as done in DETReg or UP-DETR, introduces a discrepancy in the information contained in the features (Figure 1, *c.*). Indeed, while the pretrained backbone has been trained to learn image-level features, the object detector must understand object-level information in the image. Aligning inconsistent features hinders the pretraining quality.

In this work, we propose *Proposal Selection Contrast* (ProSeCo), an unsupervised pretraining method using transformer-based detectors with a fixed pretrained backbone. ProSeCo makes use of two models. The first one aims to alleviate the discrepancy in the features by maintaining a copy of the whole detection model. This model is referred to as a *teacher* in charge of the *object proposals* embeddings, and is updated through an Exponential Moving Average (EMA) of another *student* network making the object predictions and using a similar architecture. This latter network is trained by a contrastive learning approach leveraging the high number of object proposals that can be obtained from the detectors. This methodology, in addition to the absence of batch normalization in the architectures, reduces the need for a large batch size. We further adapt the contrastive loss commonly used in pretraining to take into account the locations of the object proposals in the image, which is crucial in object detection. In addition, the localization task is independently learned through region proposals generated by Selective Search (Uijlings et al., 2013). Our contributions are summarized as:

- We propose *Proposal Selection Contrast (ProSeCo)*, a contrastive learning method tailored for pretraining transformer-based object detectors.
- We introduce the information of the localization of object proposals for the selection of positive examples in the contrastive loss to improve its efficiency for pretraining.
- We show that our proposed ProSeCo outperforms previous pretraining methods for transformer-based object detectors on standard benchmarks as well as novel benchmarks.

## 2 RELATED WORK

**Supervised Object Detection with transformer-based architectures** Object Detection is an important and extensively researched problem in computer vision (Girshick et al., 2014; Girshick,

2015; Ren et al., 2015; Redmon et al., 2016; Liu et al., 2016; Lin et al., 2017; Tian et al., 2019). It essentially combines object localization and classification tasks. Recently, a novel detector based on an encoder-decoder architecture using transformers (Vaswani et al., 2017) has been proposed in Carion et al. (2020). The training complexity of this architecture was subsequently improved in Deformable DETR (Def. DETR) (Zhu et al., 2021), by changing the attention operations into deformable attention, resulting in a faster convergence speed. Several other follow-up works (Dai et al., 2021a; Meng et al., 2021; Wang et al., 2021b; Liu et al., 2022a; Yang et al., 2022; Li et al., 2022) have also focused on increasing the training efficiency of DETR. Transformer-based architectures now represent a strong alternative to traditional convolutional object detectors, reaching better performance for a similar training cost. Furthermore, recent work have shown strong results of transformer-based detectors in a data-scarce setting (Bar et al., 2022; Bouniot et al., 2023), compared to convolutional architectures (Liu et al., 2020; 2022b), which we also observe and discuss in Appendix H.

**Self-supervised and unsupervised pretraining backbone architectures**  Recent advances in self-supervised pretraining (Grill et al., 2020; Caron et al., 2020; Chen et al., 2020b; 2021; Zheng et al., 2021; Denize et al., 2023) have achieved strong results for obtaining general representations that transfer well to image classification (He et al., 2020). Early works proposed pretext tasks (Alexey et al., 2015; Noroozi & Favaro, 2016; Komodakis & Gidaris, 2018), which are now outperformed by *Contrastive Learning* (Oord et al., 2018; Wu et al., 2018; He et al., 2020; Chen et al., 2020a; Misra & Maaten, 2020; Grill et al., 2020; Caron et al., 2020; Chen et al., 2020b; 2021; Denize et al., 2023). This paradigm relies on instance discrimination using a pair of positive views from the same input contrasted with all other instances in the batch, called negatives. However, the InfoNCE objective function (Oord et al., 2018) widely used for contrastive learning and its recent improved version, SCE (Denize et al., 2023), both require a large amount of negative instances to be effective (Wang & Isola, 2020). The improvements observed using a general self-supervised pretraining are less significant for more complex, dense downstream tasks (He et al., 2019; Reed et al., 2022). To address this issue, recent approaches have started investigating pretraining approaches tailored for these tasks by imposing local consistency, either at the *pixel* or *region-level*: they respectively propose to match in the representation space the features corresponding to the same location in the input space (O Pinheiro et al., 2020; Xie et al., 2021; Wang et al., 2021a), or apply local consistency between features from regions in the image (Roh et al., 2021; Yang et al., 2021; Xiao et al., 2021).

**Unsupervised Pretraining for the overall detection model**  Few approaches in the literature have tackled the problem of pretraining the detection-specific parts of the architecture, along with the backbone (Wei et al., 2021), or independently (Dai et al., 2021b; Bar et al., 2022). SoCo (Wei et al., 2021) proposes a pretraining strategy for convolutional detectors inspired by BYOL (Grill et al., 2020). Object locations are generated using Selective Search (Uijlings et al., 2013), and then object-level features are extracted and contrasted with each other using a *teacher-student* strategy. The small amount of object proposals and object features generated requires using a large batch size for the contrastive loss to be effective. Pretraining the backbone along with the detection modules this way makes the method difficult and costly to train due to the high amount of parameters to learn. For transformer-based architectures, UP-DETR (Dai et al., 2021b) and DETReg (Bar et al., 2022) use a fixed pretrained backbone to extract features respectively from random patch, or from object locations given by Selective Search (Uijlings et al., 2013), then pretrain the detector by localizing and reconstructing the features of the patch extracted from the input images. However, since the features to reconstruct are obtained by a backbone which was trained to encode image-level information, there is a discrepancy in the information between the features to match.

Our proposed ProSeCo is designed specifically for transformer-based detectors, and use a fixed backbone pretrained for *local information*. In this work, we leverage the high amount of object proposals generated by transformer-based detectors as instances for contrastive learning. Target object-level features and localizations are provided by a *teacher* detection model updated through EMA, inspired by recent advances in self-supervised and semi-supervised learning (Liu et al., 2020; Grill et al., 2020; Denize et al., 2023; Wei et al., 2021). The *student* detection model is pretrained by computing the contrastive loss between *object proposals* given by the student and teacher detectors. The large number of proposals generated by transformer-based detectors alleviates the need for a large batch size for the contrastive loss. The contrastive loss function used is further improved by introducing the location of objects for selecting positive proposals.

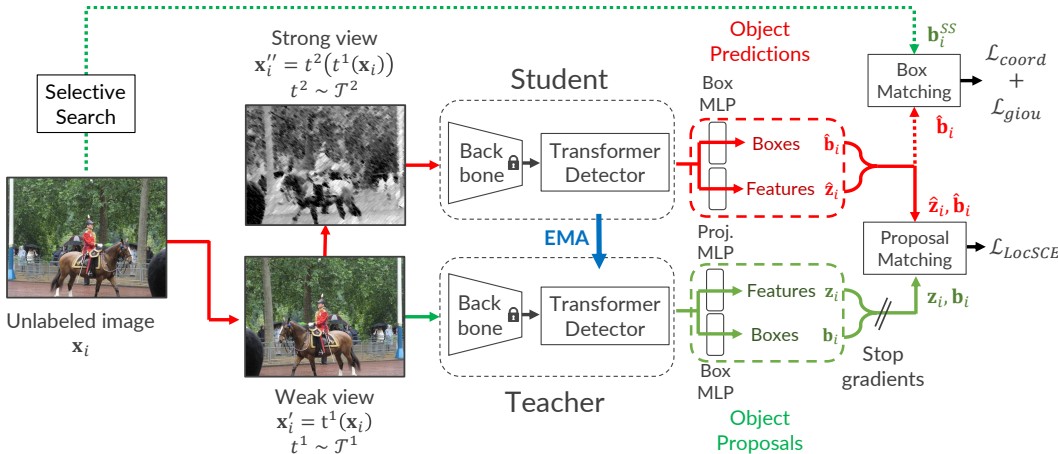

Figure 2: Overview of our proposed *ProSeCo* for unsupervised pretraining. The method follows a *student-teacher architecture*, with the teacher updated through an Exponential Moving Average (EMA) of the student. For each input image, $K$ random boxes are computed using the *Selective Search* algorithm, and two different views are generated through an asymmetric set of *weak augmentations* $\mathcal{T}^1$ and *strong augmentations* $\mathcal{T}^2$. Then, *object predictions* are obtained from the student model for the strongly augmented view, and *object proposals* from the teacher model with the weakly augmented view. Finally, the boxes predicted by the student are matched with the boxes sampled from Selective Search to compute the localization losses $\mathcal{L}_{coord}$ and $\mathcal{L}_{giou}$, and the full predictions are matched with the object proposals to compute *our novel contrastive loss* $\mathcal{L}_{LocSCE}$.

## 3 OVERVIEW OF THE APPROACH

We present in this section our proposed unsupervised pretraining approach, illustrated in Figure 2, beginning with the data processing pipeline. Then, we detail the contrastive loss used with the localization-aware positive object proposal selection. The transformer-based detectors are built on a general architecture that consists of a backbone encoder (*e.g.* a ResNet-50), followed by several transformers encoder and decoder layers, and finally two prediction heads for the bounding boxes coordinates and class logits (Carion et al., 2020; Zhu et al., 2021).

### 3.1 DATA PROCESSING PIPELINE

Throughout the section, we assume to have sampled a batch of unlabeled images $\mathbf{x} = \{\mathbf{x}_i\}_{i=1}^{N_b}$, with $\mathbf{x}_i$ the $i^{\text{th}}$ image and $N_b$ the batch size.

For each input image $\mathbf{x}_i$, we compute two different views with two asymmetric distributions of augmentations $\mathcal{T}^1$ and $\mathcal{T}^2$: a *weakly augmented view* $\mathbf{x}_i' = t^1(\mathbf{x}_i)$, with $t^1 \sim \mathcal{T}^1$, and a *strongly augmented view* obtained from the weakly augmented one $\mathbf{x}_i'' = t^2(t^1(\mathbf{x}_i))$, with $t^2 \sim \mathcal{T}^2$.

To guide the model into discovering localization of objects in unlabeled images and prevent collapse, we use bounding boxes obtained from the *Selective Search* algorithm (Uijlings et al., 2013), similarly to previous work (Wei et al., 2021; Bar et al., 2022). Since Selective Search is deterministic and the generated boxes are not ordered, we compute the boxes for all images offline and, at training time, *randomly* sample $K$ boxes $\mathbf{b}_i^{SS} = \{\mathbf{b}_{(i,j)}^{SS} \in \mathbb{R}^4\}_{j=1}^K$ for each image in the batch. Then, the two views and the corresponding boxes sampled are used as input for our method.

### 3.2 PRETRAINING METHOD

As shown in Figure 2, our approach is composed of a *student-teacher architecture* (Grill et al., 2020; Denize et al., 2023; Wei et al., 2021). With ProSeCo, we extend the student-teacher pretraining for transformer-based detectors to tackle the discrepancy in information-level when aligning features, and introduce a *dual unsupervised bipartite matching* presented below.

First, the backbone and the detection heads are respectively initialized from pretrained weights and randomly for both the student and teacher models. The teacher model is updated through an Exponential Moving Average (EMA) of the student's weights at every training iteration. For both models, the classification heads in the detectors are replaced by an MLP, called *projector*, for obtaining latent representations of the objects.

From the weakly augmented view $\mathbf{x}_i'$, the teacher model provides *object proposals* $\mathbf{y}_i = \{\mathbf{y}_{(i,j)}\}_{j=1}^N = \{(\mathbf{z}_{(i,j)}, \mathbf{b}_{(i,j)}\}_{j=1}^N$, with $\mathbf{z}_{(i,j)}$ the latent embedding and $\mathbf{b}_{(i,j)}$ the coordinates of the $j^{\text{th}}$ object found. The student model infers predictions $\hat{\mathbf{y}}_i = \{\hat{\mathbf{y}}_{(i,j)}\}_{j=1}^N = \{(\hat{\mathbf{z}}_{(i,j)}, \hat{\mathbf{b}}_{(i,j)}\}_{j=1}^N$ from the corresponding strongly augmented view $\mathbf{x}_i''$.

Then, we apply an *unsupervised* Hungarian algorithm for *proposal matching* to find from all the permutations of $N$ elements $\mathfrak{S}_N$, the optimal bipartite matching $\hat{\sigma}_i^{\text{prop}}$ between the predictions $\hat{\mathbf{y}}_i$ of the student and the object proposals $\mathbf{y}_i$ of the teacher:

$$\hat{\sigma}_i^{\text{prop}} = \arg \min_{\sigma \in \mathfrak{S}_N} \sum_{j=1}^N \mathcal{L}_{\text{prop\_match}}(\mathbf{y}_{(i,j)}, \hat{\mathbf{y}}_{(i,\sigma(j))}). \tag{1}$$

Therefore, for each image $\mathbf{x}_i$, the $j^{\text{th}}$ proposal $\mathbf{y}_{(i,j)}$ found by the teacher is associated to the $\hat{\sigma}_i^{\text{prop}}(j)^{\text{th}}$ prediction of the student $\hat{\mathbf{y}}_{(i,\hat{\sigma}_i^{\text{prop}}(j))}$. Our matching cost $\mathcal{L}_{\text{prop\_match}}$ for the Hungarian algorithm takes into account both features and bounding box predictions through a linear combination of features similarity $\mathcal{L}_{\text{sim}}(\mathbf{z}_{(i,j)}, \hat{\mathbf{z}}_{(i,\sigma(j))}) = \frac{\langle \mathbf{z}_{(i,j)}, \hat{\mathbf{z}}_{(i,\sigma(j))} \rangle}{\|\mathbf{z}_{(i,j)}\|_2 \cdot \|\hat{\mathbf{z}}_{(i,\sigma(j))}\|_2}$, the $\ell_1$ loss of the box coordinates $\mathcal{L}_{\text{coord}} = \|\mathbf{b}_{(i,j)} - \hat{\mathbf{b}}_{(i,\hat{\sigma}_i(j))}\|_1$, and the generalized IoU loss $\mathcal{L}_{\text{giou}}$ from Rezatofighi et al. (2019):

$$\mathcal{L}_{\text{prop\_match}}(\mathbf{y}_{(i,j)}, \hat{\mathbf{y}}_{(i,\sigma(j))}) = \Big[ \lambda_{\text{sim}} \mathcal{L}_{\text{sim}}\left(\mathbf{z}_{(i,j)}, \hat{\mathbf{z}}_{(i,\sigma(j))}\right) \\ + \lambda_{\text{coord}} \mathcal{L}_{\text{coord}}\left(\mathbf{b}_{(i,j)}, \hat{\mathbf{b}}_{(i,\sigma(j))}\right) + \lambda_{\text{giou}} \mathcal{L}_{\text{giou}}\left(\mathbf{b}_{(i,j)}, \hat{\mathbf{b}}_{(i,\sigma(j))}\right) \Big]. \tag{2}$$

Similarly, we also use an *unsupervised* Hungarian algorithm for *box matching*, to find the optimal bipartite matching $\hat{\sigma}_i^{\text{box}} \in \mathfrak{S}_N$ between the predicted boxes $\hat{\mathbf{b}}_i$ of the student and the sampled boxes $\mathbf{b}_i^{SS}$ from Selective Search, using the matching cost $\mathcal{L}_{\text{box\_match}}$:

$$\hat{\sigma}_i^{\text{box}} = \arg \min_{\sigma \in \mathfrak{S}_N} \sum_{j=1}^N \mathcal{L}_{\text{box\_match}}(\mathbf{y}_{(i,j)}, \hat{\mathbf{y}}_{(i,\sigma(j))}), \tag{3}$$

$$\mathcal{L}_{\text{box\_match}}(\mathbf{b}_{(i,j)}^{SS}, \hat{\mathbf{b}}_{(i,\sigma(j))}) = \lambda_{\text{coord}} \mathcal{L}_{\text{coord}}\left(\mathbf{b}_{(i,j)}^{SS}, \hat{\mathbf{b}}_{(i,\sigma(j))}\right) + \lambda_{\text{giou}} \mathcal{L}_{\text{giou}}\left(\mathbf{b}_{(i,j)}^{SS}, \hat{\mathbf{b}}_{(i,\sigma(j))}\right). \tag{4}$$

Finally, the global unsupervised loss $\mathcal{L}_u$ used for training is a combination of a loss function between the object latent embeddings of the teacher and student models, and between the object localization of the student predictions and Selective Search boxes. More formally, it is computed as:

$$\mathcal{L}_u(\mathbf{x}) = \lambda_{\text{contrast}} \mathcal{L}_{\text{LocSCE}}\left(\mathbf{y}, \hat{\mathbf{y}}, \hat{\sigma}^{\text{prop}}\right) + \\ \frac{1}{N_b K} \sum_{i=1}^{N_b} \Big[ \sum_{j=1}^K \lambda_{\text{coord}} \mathcal{L}_{\text{coord}}\left(\mathbf{b}_{(i,j)}^{SS}, \hat{\mathbf{b}}_{(i,\hat{\sigma}_i^{\text{box}}(j))}\right) + \sum_{j=1}^K \lambda_{\text{giou}} \mathcal{L}_{\text{giou}}\left(\mathbf{b}_{(i,j)}^{SS}, \hat{\mathbf{b}}_{(i,\hat{\sigma}_i^{\text{box}}(j))}\right) \Big]. \tag{5}$$

In the above equations, we define $\lambda_{\text{sim}}, \lambda_{\text{coord}}, \lambda_{\text{giou}}, \lambda_{\text{contrast}} \in \mathbb{R}$ as the coefficients for the different losses. For the consistency in the latent embeddings of the objects, we introduce the object locations information in our contrastive loss $\mathcal{L}_{\text{LocSCE}}$. This loss is used to contrast the predictions $\hat{\mathbf{y}} = \{\hat{\mathbf{y}}_i\}_{i=1}^{N_b}$ of the student with object proposals $\mathbf{y} = \{\mathbf{y}_i\}_{i=1}^{N_b}$ found by the teacher, matched according to the proposal matching $\hat{\sigma}^{\text{prop}} = \{\hat{\sigma}_i^{\text{prop}}\}_{i=1}^{N_b}$ over the batch. We detail the computations behind this loss in the next section.

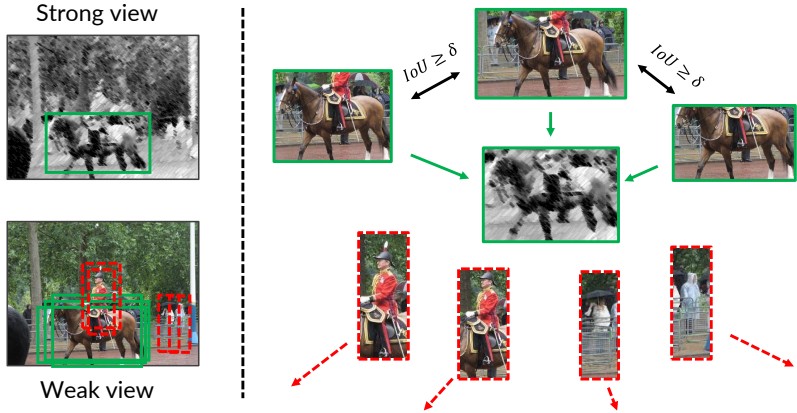

Figure 3: Illustration of the effect of the *localized contrastive loss* used. Predictions of the student and teacher models are contrasted with each other to leverage the large number of object proposals obtained from transformer-based detectors. To introduce the *object locations* information, overlapping proposals (*in green*) in each weak view, according to an *IoU threshold* $\delta$, are also considered as positive along with the matched proposal. Proposals that neither match nor overlap the matched proposal, are considered as negative (*in red*) in the contrastive loss.

### 3.3 LOCALIZATION-AWARE CONTRASTIVE LOSS

Inspired by advances in self-supervised learning, we propose a contrastive objective function (Oord et al., 2018; Chen et al., 2020a; He et al., 2020) between object proposals that also maintains relations (Zheng et al., 2021; Denize et al., 2023) among these proposals. This objective function extends the latest SCE (Denize et al., 2023) for *instance discrimination between object proposals* and is illustrated in Figure 3. We compute the contrastive loss between all the object latent embeddings from a batch of image. The *positive pair* of objects in an image $\mathbf{x}_i$ is given by the proposal matching $\sigma_i^{\text{prop}}$. First, we define the distributions of similarity between objects embeddings :

$$p'_{(in,jm)} = \frac{\mathbb{1}_{i \neq n}\mathbb{1}_{j \neq m}\exp(\mathbf{z}_{(i,j)} \cdot \mathbf{z}_{(n,m)}/\tau_t)}{\sum_{k=1}^{N_b}\sum_{l=1}^{N}\mathbb{1}_{i \neq k}\mathbb{1}_{j \neq l}\exp(\mathbf{z}_{(i,j)} \cdot \mathbf{z}_{(k,l)}/\tau_t)}, \tag{6}$$

$$p''_{(in,jm)} = \frac{\exp(\mathbf{z}_{(i,j)} \cdot \hat{\mathbf{z}}_{(n,m)}/\tau)}{\sum_{k=1}^{N_b}\sum_{l=1}^{N}\exp(\mathbf{z}_{(i,j)} \cdot \hat{\mathbf{z}}_{(k,l)}/\tau)}. \tag{7}$$

The distribution $p'_{(in,jm)}$ represents the *relations* between weakly augmented object embeddings scaled by the temperature $\tau_t$, and $p''_{(in,jm)}$ the similarity between the strongly augmented embeddings and the weakly augmented ones, scaled by $\tau$. Then, the *target similarity distribution* for the objective function is a weighted combination of one-hot label and the teacher's embeddings relations:

$$w_{(in,jm)} = \lambda_{\text{SCE}} \cdot \mathbb{1}_{i=n}\mathbb{1}_{j=m} + (1 - \lambda_{\text{SCE}}) \cdot p'_{(in,jm)}. \tag{8}$$

To introduce the localization information in our objective, we compute the pairwise *Intersection over Union* (IoU) between object proposals of the same image and consider overlapping objects as other positives when computing the target similarity distribution:

$$w^{\text{Loc}}_{(in,jm)} = \lambda_{\text{SCE}} \cdot \mathbb{1}_{i=n}\mathbb{1}_{IoU_i(j,m) \geq \delta} + (1 - \lambda_{\text{SCE}}) \cdot p'_{(in,jm)}, \tag{9}$$

where $IoU_i(j,m)$ corresponds to the IoU between teacher proposals $\mathbf{y}_{(i,j)}$ and $\mathbf{y}_{(i,m)}$ found in the same image $\mathbf{x}_i$, and $\delta$ is the *IoU threshold* to consider the proposal as a positive example. Finally, we use this tailored target similarity distribution in our *Localized SCE* (LocSCE) loss:

$$\mathcal{L}_{\text{LocSCE}}(\mathbf{y}, \hat{\mathbf{y}}, \hat{\sigma}^{\text{prop}}) = -\frac{1}{N_b N}\sum_{i=1}^{N_b}\sum_{n=1}^{N_b}\sum_{j=1}^{N}\sum_{m=1}^{N} w^{\text{Loc}}_{(in,jm)}\log(p''_{(in,j\hat{\sigma}_n^{\text{prop}}(m))}). \tag{10}$$

Note that we do not match the proposals according to $\hat{\sigma}^{\mathrm{prop}}$ in the target similarity, as we compare proposals obtained by the teacher model. We also require the full proposal as input of the loss to compute the pairwise IoU using the box coordinates. Furthermore, we recover the original formulation of SCE when $\delta = 1$. This formulation leads to an effective batch size of $N_b \cdot N$.

This localization-aware contrastive loss function aims to pull together the objects embeddings that overlap subsequently with each others, as they should correspond to the same object in the image.

## 4 EXPERIMENTS

In this section, we present a comparative study of the results of our proposed method on standard and novel benchmarks for learning with fewer data, as well as an ablative study on the most relevant parts. First, we introduce the datasets, evaluation and training settings.

### 4.1 IMPLEMENTATION DETAILS

**Datasets and evaluation**  We use *ImageNet ILSVRC 2012 (IN)* (Russakovsky et al., 2015) for pretraining, *MS-COCO (COCO)* (Lin et al., 2014) and *Pascal VOC 2007 and 2012* (Everingham et al., 2010) for finetuning. To evaluate the performance in learning with fewer data, following previous work (Wei et al., 2021; Bar et al., 2022), we consider the *Mini-COCO* benchmarks, where we randomly sample 1%, 5% or 10% of the training data. Similarly, we also introduce the novel *Mini-VOC* benchmark, in which we randomly sample 5% or 10% of the training data. We also use the Few-Shot Object Detection (FSOD) dataset (Fan et al., 2020) in the novel *FSOD-test* and *FSOD-train* benchmarks. We separate the *FSOD test set* with 80% of the data randomly sampled for training and the remaining 20% data for testing, by taking care of having at least 1 image for each class in both subsets, and do the same for the *FSOD train set*. In all benchmarks, the image ids selected for training and testing will be made available for reproducibility. More details in Appendix A.

**Pretraining**  We initialize the backbone with the publicly available pretrained SCRL (Roh et al., 2021) checkpoint and pretrain ProSeCo for 10 epochs on IN. The hyperparameters are set as follows: the EMA keep rate parameter to 0.999, the IoU threshold $\delta = 0.5$, a batch size of $N_b = 64$ images over 8 A100 GPUs, and the coefficients in the different losses $\lambda_{\mathrm{sim}} = \lambda_{\mathrm{contrast}} = 2$ which is the same value used for the coefficient governing the class cross-entropy in the supervised loss. The projector is defined as a 2-layer MLP with a hidden layer of 4096 and a last layer of 256, without batch normalization. Following SCE (Denize et al., 2023), we set the temperatures $\tau = 0.1, \tau_t = 0.07$ and the coefficient $\lambda_{\mathrm{SCE}} = 0.5$. We sample $K = 30$ *random* boxes from the outputs of Selective Search for each image at every iteration. Other training and architecture hyperparameters are defined as in Def. DETR (Zhu et al., 2021) with, specifically, the coefficients $\lambda_{\mathrm{coord}} = 5$ and $\lambda_{\mathrm{giou}} = 2$, the number of object proposals (queries) $N = 300$, and the learning rate is set to $lr = 2 \cdot 10^{-4}$. For weak augmentations $\mathcal{T}^1$, we use a random combination of flip, resize and crop, and for strong augmentations $\mathcal{T}^2$, we use a random combination of color jittering, grayscale and Gaussian blur. In $\mathcal{T}^1$, we resize images with the same range of scales as the supervised training protocol on COCO (*Large-scale*). The exact parameters for the augmentations are detailed in Appendix B, and a discussion about the pretraining cost can be found in Appendix C.

**Finetuning protocols**  For finetuning the pretrained models, we follow the standard supervised learning hyperparameters of Def. DETR (Zhu et al., 2021). In all experiments, we train the models with a batch size of 32 images over 8 A100 GPUs until the validation performance stops increasing, *i.e.* for Mini-COCO, up to 2000 epochs for 1%, 500 epochs for 5%, 400 epochs for 10%, for Mini-VOC, up to 2000 epochs for both 5% and 10%, up to 100 epochs for both FSOD-test and PASCAL VOC, and up to 50 epochs for FSOD-train. We always decay the learning rates by a factor of 0.1 after about 80% of training. Experiments with more annotated data are discussed in Appendix G. To compare our method to DETReg (Bar et al., 2022) on our novel benchmarks, we use their publicly available checkpoints from github.

### 4.2 FINETUNING AND TRANSFER LEARNING

We evaluate the transfer learning ability of our pretrained model on several datasets. Tables 1a and 1b present the results obtained compared to previous methods in the literature when learning from fewer

labeled data. We can see that our method outperforms state-of-the-art results in unsupervised pre-training on all benchmarks and datasets, and obtain even more strong results when training data is scarce. The improvement is even more significant as the overall performance with few training data is low. When using 5% of the COCO training data (*i.e.* Mini-COCO 5% in Table 1a), corresponding to about 5.9k images, ProSeCo achieves 28.8 mAP, which represents an improvement of +5.2 *percentage point* (p.p.) over the supervised pretraining baseline and +2 p.p. over both state-of-the-art overall pretraining methods. Results with all the evaluation metrics are presented in Appendix E.

Table 1: Performance (mAP in %) of our proposed pretraining approach after finetuning using different percentage of training data (with the corresponding number of images reported). We show that our ProSeCo outperforms previous pretraining methods in all benchmarks.

| Method | Detector | Pretrain. Dataset | Mini-COCO | | |
| --- | --- | --- | --- | --- | --- |
| | | | 1% (1.2k) | 5% (5.9k) | 10% (11.8k) |
| Supervised | Def. DETR | IN | 13.0 | 23.6 | 28.6 |
| SwAV (Caron et al., 2020) | Def. DETR | IN | 13.3 | 24.5 | 29.5 |
| SCRL (Roh et al., 2021) | Def. DETR | IN | 16.4 | 26.2 | 30.6 |
| DETReg (Bar et al., 2022) | Def. DETR | COCO | 15.8 | 26.7 | 30.7 |
| DETReg (Bar et al., 2022) | Def. DETR | IN | 15.9 | 26.1 | 30.9 |
| Supervised (Wei et al., 2021) | Mask R-CNN | IN | – | 19.4 | 24.7 |
| SoCo* (Wei et al., 2021) | Mask R-CNN | IN | – | 26.8 | 31.1 |
| *ProSeCo (Ours)* | Def. DETR | IN | **18.0** | **28.8** | **32.8** |

(a)

| Method | FSOD-test | FSOD-train | PASCAL VOC | Mini-VOC | |
| --- | --- | --- | --- | --- | --- |
| | 100% (11k) | 100% (42k) | 100% (16k) | 5% (0.8k) | 10% (1.6k) |
| Supervised | 39.3 | 42.6 | 59.5 | 33.9 | 40.8 |
| DETReg (Bar et al., 2022) | 43.2 | 43.3 | 63.5 | 43.1 | 48.2 |
| *ProSeCo (Ours)* | **46.6** | **47.2** | **65.1** | **46.1** | **51.3** |

(b)

## 4.3 ABLATION STUDIES

In the following, we provide several ablation studies for our proposed approach. All experiments and results are compared on the Mini-COCO 5% benchmark with the pretrained SwAV backbone unless explicitly stated. Additional ablation on the number of queries can be found in Appendix F.

**Pretraining dataset and backbone** In Table 2a, we show the effect of changing the pretraining dataset or the fixed backbone used. We can see that using a backbone more adapted to dense tasks that learned local information (*e.g.* SCRL) helps the model by having consistent features (+1 p.p.), compared to a backbone pretrained for global features (*e.g.* SwAV). Furthermore, even with a less adapted backbone, our ProSeCo initialized with the SwAV backbone outperforms DETReg (+1.7 p.p.). Pretraining with DETReg improves when using a more adapted backbone, but ProSeCo still reaches better performance. To compare with IN, we also pretrain ProSeCo on COCO for 120 epochs. We obtain better results when pretraining the model on IN than using COCO thanks to the large number of different images in IN (about 10 times the number of images of COCO, leading to +0.4 p.p.), which is consistent with previous findings (Wei et al., 2021).

**Localization information in contrastive loss** In Table 2b, we show the effect of the localization information in the contrastive loss SCE. We can see that when introducing multiple positive examples for each image based on the IoU threshold $\delta$ (*i.e.* $\forall \delta < 1$), we achieve better results than with the original SCE loss (*i.e.* $\delta = 1$). Notably, the best results are achieved with $\delta = 0.5$ (+1.7 p.p.). More experiments with the InfoNCE loss (Oord et al., 2018) can be found in Appendix D.

**Hyperparameters** Table 3 presents an ablation study on different important hyperparameters of our approach. We experimented first with the same batch size applied in Bar et al. (2022) (*Abl. Batch*),

but found that using a smaller batch size (*Base*) leads to improved results (+0.2 p.p.). We evaluated different image scales as a parameter of the weak data augmentations distribution in *Abl. Scale*. *Mid-scale* corresponds to a resizing of the images such that the shortest edge is between 320 and 480 pixels, as used in previous work (Dai et al., 2021b; Bar et al., 2022), and *Large-scale* to a resize between 480 and 800 pixels, used for supervised learning on COCO. Exact values for these parameters can be found in Appendix B. We found that increasing the size of the images during pretraining is important to have more meaningful information in the boxes, and a more precise localization of the boxes (+0.7 p.p.). Following the best results from Denize et al. (2023), we evaluated the performance for $\tau_t \in \{0.05, 0.07\}$. We found that $\tau_t = 0.07$ leads to the best performance (+0.3 p.p.). We considered several EMA keep rate parameter values following previous work (He et al., 2020; Wei et al., 2021; Denize et al., 2023), and found that 0.999 achieves the best results (+0.1 p.p.).

Table 2: (a) Comparison after finetuning when using different pretrained backbone and/or pretraining datasets. (b) Comparison of the effect of the localization information using different IoU threshold $\delta$. All performance (mAP in %) are measured on Mini-COCO 5%.

| Pretraining | Dataset | mAP |
|---|---|---|
| ProSeCo w/ SwAV | COCO | 27.4 |
| ProSeCo w/ SwAV | IN | 27.8 |
| DETReg w/ SCRL | IN | 28.0 |
| ProSeCo w/ SCRL | IN | **28.8** |

(a)

| Loss | $\delta$ | mAP |
|---|---|---|
| SCE | 1.0 | 26.1 |
| *LocSCE (Ours)* | 0.2 | 27.0 |
| *LocSCE (Ours)* | 0.7 | 27.1 |
| *LocSCE (Ours)* | 0.5 | **27.8** |

(b)

Table 3: Ablation studies on different hyperparameters for the proposed method. The performance (mAP in %) are measured on Mini-COCO 5%. **Green and bold columns names** indicate a *positive* effect on the performance, and red columns a *negative* effect.

| Ablative Variant | Batch size | | Images scale | | Temperature $\tau_t$ | | EMA | | | mAP |
|---|---|---|---|---|---|---|---|---|---|---|
| | 192 | 64 | Mid | Large | 0.05 | 0.07 | 0.99 | 0.996 | 0.999 | |
| Base | | ✓ | ✓ | | ✓ | | | ✓ | | 26.7 |
| Abl. Batch | ✓ | | ✓ | | ✓ | | | ✓ | | 26.5 |
| Abl. Scale | | ✓ | | ✓ | ✓ | | | ✓ | | 27.4 |
| Abl. Temp. | | ✓ | ✓ | | | ✓ | | ✓ | | 27 |
| Abl. EMA | | ✓ | ✓ | | ✓ | | ✓ | | | 26.3 |
| | | ✓ | ✓ | | ✓ | | | | ✓ | 26.8 |
| **Best** | | ✓ | | ✓ | | ✓ | | | ✓ | **27.8** |

## 5 CONCLUSION

In this work, we aim to use large unlabeled datasets for an unsupervised pretraining of the overall detection model to improve performance when having access to fewer labeled data. In this end, we propose *Proposal Selection Contrast (ProSeCo)*, a novel pretraining approach for Object Detection. Our method leverages the large number of object proposals generated by transformer-based detectors for contrastive learning, reducing the necessity of a large batch size, and introducing the localization information of the objects for the selection of positive examples to contrast. We show from various experiments on standard and novel benchmarks in learning with few training data that ProSeCo outperforms previous pretraining methods. Throughout this work, we advocate for consistency in the level of information encoded in the features when pretraining. Indeed, learning object-level features during pretraining is more important than image-level when applied to a dense downstream task such as Object Detection. Future work could update the backbone during pretraining to further improve the consistency between the backbone and the detection heads.

## REPRODUCIBILITY STATEMENT

Throughout this paper, we made sure that the experiments and results are fully reproducible. We explicitly state the exact values of hyperparameters used in Section 4.1, and describe in details the datasets and evaluation protocols in Appendix A. All image ids randomly selected when evaluating with few training data (in Mini-COCO, Mini-VOC, FSOD-test, FSOD-train) will be made available.

## ACKNOWLEDGEMENTS

This work was made possible by the use of the Factory-AI supercomputer, financially supported by the Ile-de-France Regional Council, and also performed using HPC resources from GENCI-IDRIS (Grant 2022-AD011013478).

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
