# OpenReview forum: "Proposal-Contrastive Pretraining for Object Detection from Fewer Data"
_ICLR.cc/2023/Conference — ICLR 2023 notable top 25%_

### Official Review · Reviewer_Sfkk · 2022-10-24

**Confidence:** 4
**Clarity, Quality, Novelty And Reproducibility:** See Strength And Weaknesses
**Correctness:** 4
**Technical Novelty And Significance:** 3
**Empirical Novelty And Significance:** 3
**Recommendation:** 6

**Strength And Weaknesses:**

Strengths
+ The contributions are clearly illustrated and validated. The ablation experiments clearly reflect these contributions. I think ProSeCo is on the right way of exploration.
+ The illustrations and explanations are clear.
+ The proposed method outperforms others by a large margin.


Weaknesses
- In the Introduction, after analyzing the local feature consistency, there is no description of its application and advantages, and the relationship with the whole approach is not clearly elaborated.
- The red part in Figure 3 is not clearly explained, which raises confusion.

**Summary Of The Paper:**

This paper proposes to solve the challenging large batch size for contrast learning in unsupervised object detection pretraining. They conducted experiments with the Transformer-based detector and found that the diversity of proposals generated by the detector can effectively mitigate this problem. Thus, they propose ProSeCo, which improves the state-of-the-art on the fewer data pretraining task. They also propose a novel localization-aware proposal matching loss and apply it in the feature-consistent student-teacher architecture, which will be very useful for object detection. I think the precision of the proposed method is obviously improved, and the pre-training for unsupervised object detection deserves more in-depth study.

**Summary Of The Review:**

See Strength And Weaknesses

---

> ### Author Response · Authors · 2022-11-11
> **Response to Reviewer**
>
> We thank the reviewer for the kind comments on the paper.
> Regarding the few weaknesses raised:
>
> > In the Introduction, after analyzing the local feature consistency, there is no description of its application and advantages, and the relationship with the whole approach is not clearly elaborated.
>
> In ProSeCo, the local feature consistency is obtained by using the *whole detection model* as a teacher. The advantages are the consistency between the information encoded at the *object level* in the object proposals of the teacher and student, compared to DETReg that has a discrepancy between the two since the teacher model is a *fixed backbone* pretrained at the *image level*. Aligning inconsistent features can hinder the pretraining quality.
> We will clarify this in the Introduction for the revision of the paper.
>
> > The red part in Figure 3 is not clearly explained, which raises confusion.
>
> In Figure 3, the red boxes correspond to *negative proposals* in the weak view according to the given proposal in the strong view. They are proposals found by the student model that *do not match* **AND** *do not overlap* with the given proposal from the teacher. We will add an explanation in the caption of Figure 3 in the revision.

---

### Official Review · Reviewer_9EDD · 2022-10-24

**Confidence:** 4
**Clarity, Quality, Novelty And Reproducibility:** The writing of the paper is clear and…
**Correctness:** 4
**Technical Novelty And Significance:** 4
**Empirical Novelty And Significance:** 3
**Recommendation:** 8

**Strength And Weaknesses:**

The paper is easy to understand and the experiment are very comprehensive. The idea of using localization information to sample proposals for pre-training is interesting to me. Overall speaking I think it's a good paper.

I have two questions of the paper: (i) what's the training time of the proposed method compared with the existing work. In addition to large batch training, the contrastive pre-training is also time-consuming; (ii) the proposed pre-training model is used for fsod, andI feel the proposed strategy is general and can also be applied in generic object detection. What's the performance for generic object detection?

**Summary Of The Paper:**

This paper proposes a novel contrastive pre-training strategy for object detection. The method uses object proposals generated by Selective Search for contrastive learning, which avoids large batch requirement in the existing work. And the localization of the proposals are considered during sampling stage, to further improve the training efficiency. The proposed strategy shows clear improvement over baselines.

**Summary Of The Review:**

Overall speaking I think it's a good paper.

---

> ### Author Response · Authors · 2022-11-11
> **Response to Reviewer**
>
> We thank the reviewer for the nice comments on the paper.
> Regarding the questions of the reviewer :
>
> > (i) what's the training time of the proposed method compared with the existing work. In addition to large batch training, the contrastive pre-training is also time-consuming;
>
> We mainly compared the cost of pretraining to SoCO (Wei et al., 2021) since it is the closest in terms of pretraining pipeline. We will clarify in the paper with a discussion in the Appendix, that we meant costly in terms of hardware and memory compared to SoCo, i.e. method e in our Figure 1, and we will change Figure 1 accordingly.
> To go more into details, from information available in their paper and official github :
> - SoCo pretrains for 400 epochs of imagenet with a total batch size of 2048 = 240k iterations in 140h = 0.5 it / sec, using *16 GPUs V100 32G*.
> - We pretrain for 10 epochs of imagenet with a total batch size of 64 = 187k iterations in 40h = 1.4 it / sec, using *8 GPUs A100 40G*.
>
> Even though the GPU A100 are faster than the V100, we are *training much faster* which is explained by the fact that they train the overall model, including the backbone, leading to more parameters to learn and thus more computations.
> Furthermore, our Proposal-Contrastive Learning requires a smaller batch size leading to less memory and thus less GPUs needed.
>
> Compared to the much simpler DETReg from their implementation details:
> - DETReg pretrains for 5 epochs of ImageNet with a total batch size of 192 = 33k iterations in 10h using *8 GPUs A100 40G*.
>
> They use smaller images and the pretraining degrades after more epochs. We assume that it comes from the second pretrained encoder backbone acting as a teacher for the features that stays fixed throughout training. In the end, they achieve lower performance.
>
> > (ii) the proposed pre-training model is used for fsod, and I feel the proposed strategy is general and can also be applied in generic object detection. What's the performance for generic object detection?
>
> That's a good question. The improvements in the large-scale annotated data regime are limited for this kind of pretraining, which can be observed also in previous work (Dai et al., 2021; Bar et al, 2022). In the following table, we present results when fine-tuning on the full COCO dataset under the $1 \times$ training schedule (Wei et al., 2021; Li et al., 2022), i.e. 12 training epochs and decaying the learning rate in the last epoch. As we can see, our ProSeCo achieves similar results as DETReg.
>
> Method | mAP (in %)
> --- | ---
> Supervised | 37.4
> DETReg | 38.9
> ProSeCo (Ours) | 38.9
>
> We believe that this limitation comes from the pretrained backbone that stays fixed during pretraining, and from the extensive supervision during fine-tuning.
> We will add this Table and discussion in the Supplementary Material in the revision.
>
> ## References :
> - Amir Bar, Xin Wang, Vadim Kantorov, Colorado J Reed, Roei Herzig, Gal Chechik, Anna Rohrbach, Trevor Darrell, and Amir Globerson. "Detreg: Unsupervised pretraining with region priors for object detection." In *Proceedings of the IEEE/CVF Conference on Computer Vision and Pattern Recognition*, 2022.
> - Fangyun Wei, Yue Gao, Zhirong Wu, Han Hu, and Stephen Lin. "Aligning pretraining for detection via object-level contrastive learning." In *Advances in Neural Information Processing Systems*, 2021
> - Feng Li, Hao Zhang, Shilong Liu, Jian Guo, Lionel M Ni, and Lei Zhang. "Dn-detr: Accelerate detr training by introducing query denoising." In *Proceedings of the IEEE/CVF Conference on Computer Vision and Pattern Recognition*, 2022
> - Zhigang Dai, Bolun Cai, Yugeng Lin, and Junying Chen. "Up-detr: Unsupervised pre-training for object detection with transformers." In *Proceedings of the IEEE/CVF conference on computer vision and pattern recognition*, 2021

---

> > ### Author Response · Authors · 2022-11-17
> > **More results in the Mid-scale annotated data regime**
> >
> > To extend the discussion about generic object detection, we also investigated the results in a setting with mid-scale annotated data, that would be between the large-scale setting with the full COCO dataset corresponding to 118k labeled data, and the small-scale setting with the different benchmarks presented in the paper (Mini-COCO, Mini-VOC, Pascal VOC and FSOD-test) with between 1k and 16k labeled data available. For this setting we propose the *FSOD-train* benchmark obtained from the training set of the FSOD dataset, that we separate between a training and testing subset similarly to the *FSOD-test* benchmark presented in the paper. This separation leads to about *42k training images* and *10k testing images*. We present the results in the following table, after fine-tuning the pretrained model on the training subset for 50 epochs:
> >
> > Method | mAP (in %)
> > --- | ---
> > Supervised | 42.6
> > DETReg | 43.3
> > ProSeCo (Ours) | **47.2**
> >
> > We can see that our proposed method is still more efficient in a setting with more annotated data. We will update the paper to add the results, the detailed experimental protocol for this benchmark, along with the exact image IDs chosen for the training and testing subsets.

---

> > > ### Comment · Reviewer_9EDD · 2022-11-28
> > > **Final Recommendation**
> > >
> > > First of all I thank the rebuttal provided by the author, which is comprehensive. My concerns have been fully resolved, including the training time and the potential usage in generic object detection. From the rebuttal, the training time is also significantly reduced and the comparable results on large scale annotation training scheme are also reported. After checking the comments from other reviewers, I think it's a good paper and keep my pre-rebuttal score.

---

### Official Review · Reviewer_cxY8 · 2022-10-25

**Confidence:** 4
**Correctness:** 2
**Technical Novelty And Significance:** 2
**Empirical Novelty And Significance:** 1
**Recommendation:** 6

**Clarity, Quality, Novelty And Reproducibility:**

The overall quality of the paper is below the acceptance threshold since there are some aspects of the paper's claims not well-supported by theoretical and experimental analysis.
The originality of the work is not significant as the contribution is limited to positive examples selection for contrastive learning.

**Strength And Weaknesses:**

Strength:
- The overall paper is well-written and defined. It is easy enough to follow and reproduce

Weaknesses:
- The reasoning for the design choice of a transformer-based detector is not well-supported. The paper's claim of the high amount of object proposals is suitable for contrastive learning. How about other types of detectors or considering low-score predictions as object proposals?
- Considering overlapping objects as other positives may not work well in other cases. How about crowd scenes?
- It is not proven the paper's claim of less costly than previous approaches.

**Summary Of The Paper:**

The paper presents an overall pretraining approach for object detection with a fixed pretrained backbone. A transformer-based detector is used.
The main contribution of the paper is the approach to selecting positive examples in a contrastive learning method.

**Summary Of The Review:**

My recommendation rating for the paper is to reject. The main reason is that some claims are not well-supported as identified in the weaknesses.

---

> ### Author Response · Authors · 2022-11-11
> **Response to Reviewer**
>
> We thank the reviewer for being positive on the overall writing of the paper.
> About the novelty and the significance of the work, we would like to point out that we are not aware of similar contributions and better results in our context with limited labels. We hope that our answers below will help to precise the level of our contribution, and will be happy to discuss the positioning of our contribution in relation to other related work.
> Regarding the weaknesses raised:
>
> ### First remark
>
> We chose to focus on transformer-based detector as we found that they are more suited for *learning with limited labels*, and it is consistent with experimental findings from previous work [1]. Transformer-based detectors have also reached strong results compared to convolutional detectors [5]. We mention this in the last sentence of the discussion about supervised object detection in the related work section.
> Here is a table reporting the performance (mAP in %) of Faster-RCNN (FRCNN) [2] with Feature Pyramid Network (FPN) [3], a two-stage convolutional detector commonly used for Object Detection, and Deformable DETR (Def. DETR) [6] that we mainly use in our work, from experiments on different percentages of COCO labeled training data (with the corresponding number of images), reproduced on 5 seeds to obtain different sampled data for a more detailed comparison :
>
> Method | Mini-COCO 0.5% (590) | Mini-COCO 1% (1180) | Mini-COCO 5% (5900) | Mini-COCO 10% (11800) |
> --- | --- | --- | --- | --- |
> FRCNN + FPN | $6.83 \pm 0.15$ | $9.05 \pm 0.16$ | $18.47 \pm 0.22$ | $23.86 \pm 0.81$|
> Def. DETR | $\mathbf{8.95 \pm 0.51}$ | $\mathbf{12.96 \pm 0.08}$ | $\mathbf{23.59 \pm 0.21}$ | $\mathbf{28.55 \pm 0.08}$ |
>
> From the results presented in the table, we can see that Def. DETR achieves consistently better performance than the most popular two-stage method when learning with limited labels. These results also motivated our choice of transformer-based architectures for our pretraining method.
> We will add this table in the supplementary materials and refer to it to further motivate our choice.
> Using our proposed LocSCE with other types of detector would be an interesting investigation, but it would require a whole different analysis and is currently out of the scope of this paper.  Furthermore, [4] have analyzed the optimal number of proposals for their pretraining approach SoCo [4] that uses a Mask R-CNN, and found that *only 4 proposals per image* is better in their case, which is much smaller than the 300 proposals of Def. DETR. This also explains why they require large batch size for pretraining.
>
> ### Second remark
>
> We only consider overlapping objects as other positives *during the unsupervised pretraining phase* using our proposed LocSCE. The fine-tuning part afterwards follows the classic supervised learning procedure.
> Crowd scenes might be less suited *for pretraining* compared to ImageNet due to a lower diversity in images. However, as long as we can learn an object detector on the crowd scene dataset, there should be no problem during the fine-tuning phase.
> Note that the IoU threshold can be tuned, we found that 0.5 works best for pretraining on ImageNet, but maybe a higher threshold would be better for a more crowded pretraining dataset. However, an IoU of 0.5 is already high since it means that about 2/3 of the box is covered by the other one.
>
> ### Third remark
>
> We will clarify in the paper that we meant costly in terms of hardware and memory compared to SoCo [4], i.e. method e in our Figure 1, and we will change Figure 1 accordingly.
> However, to prove our claim and go more into details, from information available in their paper and official github :
> - SoCo pretrains for 400 epochs of imagenet with a total batch size of 2048 = 240k iterations in 140h = 0.5 it / sec, using *16 GPUs V100 32G*.
> - We pretrain for 10 epochs of imagenet with a total batch size of 64 = 187k iterations in 40h = 1.4 it / sec, using *8 GPUs A100 40G*.
>
> Even though the GPU A100 are faster than the V100, we are *training much faster* which is explained by the fact that they train the overall model, including the backbone, leading to more parameters to learn and thus more computations.
> Furthermore, our Proposal-Contrastive Learning requires a smaller batch size leading to less memory and thus less GPUs needed.
>
> ## References :
> - [1] Bar et al., "Detreg: Unsupervised pretraining with region priors for object detection." *CVPR*, 2022.
> - [2] Ren et al., "Faster r-cnn: Towards real-time object detection with region proposal networks." *NeurIPS*, 2015.
> - [3] Lin et al., "Feature pyramid networks for object detection." *CVPR*, 2017.
> - [4] Wei et al., "Aligning pretraining for detection via object-level contrastive learning." *NeurIPS*, 2021
> - [5] Li et al., "Dn-detr: Accelerate detr training by introducing query denoising." *CVPR*, 2022
> - [6] Zhu et al., "Deformable detr: Deformable transformers for end-to-end object detection." *ICLR*, 2021.

---

> > ### Comment · Reviewer_cxY8 · 2022-12-01
> > **Re: Response to Reviewer**
> >
> > Thank you for the feedback provided by the authors. Most of my concerns are addressed in their second and third remark. For a better picture, I recommend the author compare with another popular one-stage detector approach, e.g. CenterNet, YOLOX, or a more recent one. FRCNN +FPN is outdated (since 2015, 2017) compared with Def DETR (since 2021).

---

> > > ### Author Response · Authors · 2022-12-01
> > > **Comparison with one-stage detector from recent work**
> > >
> > > We thank the reviewer for taking the time to read our feedback. At this stage, we cannot update the paper with more experiments, but we agree it would be interesting to compare results on few labeled data with other detectors.
> > > First, we would like to note that even though Faster RCNN was released some years ago, it is still used as a strong baseline in recent papers, even in the original paper of Deformable DETR [6]. Furthermore, SoCo [4] is implemented with Mask RCNN [7] which is based on Faster RCNN, so it also makes sense in our case to compare Deformable DETR with Faster RCNN.
> > > Nevertheless, recent work [8] provided experiments with both FCOS [9], a more recent one-stage detector, and Faster RCNN on different fractions of the COCO training set (in Table 4 and Table 5 of [8]). We can use these results to update our above table :
> > >
> > > Method | Mini-COCO 0.5% (590) | Mini-COCO 1% (1180) | Mini-COCO 5% (5900) | Mini-COCO 10% (11800) |
> > > --- | --- | --- | --- | --- |
> > > FCOS | $5.42 \pm 0.01$ | $8.43 \pm 0.03$ | $17.01 \pm 0.01$ | $20.98 \pm 0.01$
> > > FRCNN + FPN | $6.83 \pm 0.15$ | $9.05 \pm 0.16$ | $18.47 \pm 0.22$ | $23.86 \pm 0.81$|
> > > Def. DETR | $\mathbf{8.95 \pm 0.51}$ | $\mathbf{12.96 \pm 0.08}$ | $\mathbf{23.59 \pm 0.21}$ | $\mathbf{28.55 \pm 0.08}$ |
> > >
> > > With these updated results, we can see that Deformable DETR remains a strong choice with few labeled data.
> > >
> > > ### References
> > >
> > > - [4] Wei et al., "Aligning pretraining for detection via object-level contrastive learning." *NeurIPS*, 2021
> > > - [6] Zhu et al., "Deformable detr: Deformable transformers for end-to-end object detection." *ICLR*, 2021.
> > > - [7] He et al., "Mask R-CNN", *ICCV*, 2017
> > > - [8] Liu et al., "Unbiased Teacher v2: Semi-supervised Object Detection for Anchor-free and Anchor-based Detectors", *CVPR*, 2022
> > > - [9] Tian et al., "FCOS: Fully convolutional one-stage object detection", *ICCV*, 2019

---

### Author Response · Authors · 2022-11-11
**Revision of the paper**

We thank all the reviewers for their helpful comments. We revised the paper to include all the comments, the additions in the text of the paper are written in blue. Here is a summary of the revision:
- We updated Figure 1 to more clearly show that we compare the cost of pretraining with method e. We also added a discussion in the Appendix C with a detailed comparison in terms of memory and hardware used to SoCo. *(reviewers cxY8 and 9EDD)*
- We added a comment in the Introduction to explain the problem of having a discrepancy in the features between encoder and detection heads. *(reviewer Sfkk)*
- We discuss in the Appendix H the difference of performance observed between convolutional and transformer-based detectors in the context of training with limited labels. *(reviewer cxY8)*
- We clarify the red boxes in Figure 3. *(reviewer Sfkk)*
- We present experiments with large-scale annotated data in the Appendix G. *(reviewer 9EDD)*

---

### Decision · Program_Chairs · 2023-01-20

**Decision:**

Accept: notable-top-25%

**Justification For Why Not Higher Score:**

To me the paper was between Accept (poster) and Accept (spotlight). I've decided to go with a more optimistic recommendation.

**Justification For Why Not Lower Score:**

Paper is well written and proposes both interesting and practical approach for an important problem.

**Metareview: Summary, Strengths And Weaknesses:**

Paper was reviewed by three reviewers. While initial reviews were not as positive, after the rebuttal, all reviewers are arguing for acceptance with: 1 x accept, good paper and 2 x marginally above the acceptance threshold. Concerns of both cxY8 and 9EDD were appropriately and fully addressed; Sfkk has not responded to the rebuttal. Overall, reviewers agree that the paper is well written, contributions are clear and performance is substantially better than the baselines. AC has looked at reviews, rebuttal and following discussion and concurs with reviewers that paper would make a valuable contribution to the conference. Authors are encouraged to address remaining comments (e.g., from cxY8 with respect to additional comparisons) for the camera ready.

**Note From Pc:**

if the above contains the word "oral" or "spotlight" please see: "oral" presentation means -> notable-top-5% and "spotlight" means -> notable-top-25%. As stated in our emails, we are disassociating presentation type from AC recommendations